# Improving the Solubility and Oral Bioavailability of a Novel Aromatic Aldehyde Antisickling Agent (PP10) for the Treatment of Sickle Cell Disease

**DOI:** 10.3390/pharmaceutics13081148

**Published:** 2021-07-27

**Authors:** Tarek A. Ahmed, Khalid M. El-Say, Fathy I. Abd-Allah, Abdelsattar M. Omar, Moustafa E. El-Araby, Yosra A. Muhammad, Piyusha P. Pagare, Yan Zhang, Khadijah A. Mohmmad, Osheiza Abdulmalik, Martin K. Safo

**Affiliations:** 1Department of Pharmaceutics, Faculty of Pharmacy, King Abdulaziz University, Jeddah 21589, Saudi Arabia; kelsay1@kau.edu.sa; 2Department of Pharmaceutics and Industrial Pharmacy, Faculty of Pharmacy, Al-Azhar University, Cairo 11884, Egypt; fibrahim@icbr.info; 3Department of Pharmaceutical Chemistry, Faculty of Pharmacy, King Abdulaziz University, Jeddah 21589, Saudi Arabia; asmansour@kau.edu.sa (A.M.O.); madaoud@kau.edu.sa (M.E.E.-A.); ymuhammad@kau.edu.sa (Y.A.M.); kmohammad@kau.edu.sa (K.A.M.); 4Department of Pharmaceutical Chemistry, Faculty of Pharmacy, Al-Azhar University, Cairo 11884, Egypt; 5Department of Medicinal Chemistry, Virginia Commonwealth University, Richmond, VA 23298, USA; pagarepp@vcu.edu (P.P.P.); yzhang2@vcu.edu (Y.Z.); msafo@vcu.edu (M.K.S.); 6Division of Hematology, The Children’s Hospital of Philadelphia, Philadelphia, PA 19104, USA; abdulmalik@chop.edu; 7Development, School of Pharmacy, The Institute for Structural Biology, Drug Discovery, Virginia Commonwealth University, Richmond, VA 23298, USA

**Keywords:** PP10, aromatic aldehyde, sickle cell disease, inherited blood disorders, oral tablets, intravenous formulation, bioavailability

## Abstract

Background: Aromatic aldehydes, with their ability to increase the oxygen affinity of sickle hemoglobin, have become important therapeutic agents for sickle cell disease (SCD). One such compound, voxelotor, was recently approved for SCD treatment. Methyl 6-((2-formyl-3-hydroxyphenoxy)methyl) picolinate (PP10) is another promising aromatic aldehyde, recently reported by our group. Like voxelotor, PP10 exhibits O_2_-dependent antisickling activity, but, unlike voxelotor, PP10 shows unique O_2_-independent antisickling effect. PP10, however, has limited solubility. This study therefore aimed to develop oral and parenteral formulations to improve PP10 solubility and bioavailability. Methods: Oral drug tablets with 2-hydroxypropyl beta cyclodextrin (HP-β-CD), polyvinylpyrrolidone, or Eudragit L100-55 PP10-binary system, and an intravenous (IV) formulation with d-α-tocopherol polyethylene glycol 1000 succinate (TPGS) or HP-β-CD, were developed. The pharmacokinetic behavior of the formulations was studied in Sprague-Dawley rats. PP10, a methylester, and its acid metabolite were also studied in vitro with sickle whole blood to determine their effect on Hb modification, Hb oxygen affinity, and sickle red blood cell inhibition. Results: Aqueous solubility of PP10 was enhanced ~5 times with the HP-β-CD binary system, while the TPGS aqueous micelle formulation was superior, with a drug concentration of 0.502 ± 0.01 mg/mL and a particle size of 26 ± 3 nm. The oral tablets showed relative and absolute bioavailabilities of 173.4% and 106.34%, respectively. The acid form of PP10 appeared to dominate in vivo, although both PP10 forms demonstrated pharmacologic effect. Conclusion: Oral and IV formulations of PP10 were successfully developed using HP-β-CD binary system and TPGS aqueous micelles, respectively, resulting in significantly improved solubility and bioavailability.

## 1. Introduction

Sickle cell disease is an inherited blood disorder that affects millions of people worldwide [1,2]. A single-point mutation (βGlu6→βVal6) of hemoglobin (Hb), the protein that transports oxygen (O_2_) from the lungs to tissues, leads to formation of sickle Hb (HbS). Under hypoxia, and as the concentration of the low O_2_-affinity deoxygenated HbS increases, the protein begins to polymerize into fibers due to interactions between the mutated βVal6 residue and a hydrophobic pocket on an adjacent HbS tetramer, causing sickling of red blood cells (RBCs). These rigid and brittle RBCs impair blood flow, causing hemolysis and vaso-occlusion (VOC), which leads to other downstream adverse effects, e.g., adhesion of RBCs to tissue endothelium, hemolysis, oxidative stress, decreased vascular nitric oxide (NO) bioavailability, inflammation, painful VOC crisis, and, eventually, chronic endothelial and organ damage that ultimately leads to poor quality of life and decreased life expectancy [1,2,3,4,5]. Four drugs are currently approved by the FDA to treat the disease. These include hydroxyurea, which has been used for more than two decades [6,7], and L-glutamine (Endari) [8,9], crizanlizumab (Adakveo) [10], and voxelotor (Oxbryta, otherwise known as GBT440) [11], which were approved over the last four years. Voxelotor, the latest drug to be approved, is an aromatic aldehyde that prevents hypoxia-induced HbS polymerization and RBC sickling by increasing the oxygen affinity of HbS [12,13,14].

Dissolution rate is considered to be the rate-limiting step in the absorption process of a drug in solid dosage form. Poor dissolution profile of water insoluble drugs has been, and still remains, a great problem in the development of effective medications [15]. Hence, different strategies are employed to enhance the dissolution of poorly water-soluble drugs. Solid dispersion is among these strategies, which involves dispersion of a hydrophobic drug in at least one hydrophilic carrier. The process results in an increase in the surface area, improving drug solubility and dissolution rate [16]. Solid dispersions can be prepared utilizing different methods, such as kneading, solvent evaporation, melting, and melting-solvent techniques. Spray drying, lyophilization, supercritical fluid, electrospinning, and coprecipitation are among the methods used in the solvent evaporation technique. Hot-melt extrusion and melt agglomeration are utilized in the melting/fusion method [17].

Our group, through iterative modifications of pyridine derivatives of benzaldehyde, discovered PP10 as a novel therapeutic agent for sickle cell disease [18]. PP10 binds to the α-cleft of Hb to potently increase the protein’s affinity for oxygen with a concomitant inhibition of hypoxia-induced RBC sickling in a sustained manner, both in vitro and in vivo [18]. Voxelotor, also an aromatic aldehyde, exhibits similar potent O_2_-dependent antisickling activity as PP10 [19,20,21]. However, unlike voxelotor and several other previously studied aromatic aldehydes, PP10 exhibits an additional novel O_2_-independent antisickling mechanism of action as a result of direct polymer destabilization effects [22], which are akin to that of fetal Hb (HbF) and Hb Stanleyville [23,24]. Given that SCD is an ischemia-reperfusion injury syndrome with transient areas of local and/or regional hypoxia, PP10 may perform better clinically than other aromatic aldehydes by preventing HbS polymerization in areas of low oxygen tension, and at clinically safe dose levels, making SCD a highly manageable chronic condition. Unfortunately, like most therapeutically promising aromatic aldehydes, the aqueous solubility of PP10 is very poor. Specifically, the aqueous solubility of crystalline PP10 at 25 °C is 0.012 ± 0.002 mg/mL, which we were only able to marginally improve with excipients in our animal studies [18]. The poor solubility of PP10 necessitated this study’s aim to develop a proper formulation to increase its GI solubility and oral bioavailability.

In this study, oral drug tablets based on 2-hydroxypropyl beta cyclodextrin (HP-β-CD), polyvinylpyrrolidone (PVP-K90), or Eudragit L100-55 PP10-binary system, as well as an intravenous (IV) formulation with d-α-tocopherol polyethylene glycol 1000 succinate (TPGS) or 2-hydroxypropyl beta cyclodextrin (HP-β-CD), were developed and characterized. The pharmacokinetic behavior of the oral and intravenous formulations were studied in Sprague-Dawley rats. PP10 contains a methylester moiety, which could potentially hydrolyze into the corresponding PP10 acid metabolite (PP10-acid), either chemically (acid/base hydrolysis) or by esterase, in the body. Therefore, we determined the in vitro pharmacodynamic activities of PP10-acid and its sodium salt (PP10-NaSalt) and compared them with those of the parent PP10 using homozygous sickle (SS) blood.

## 2. Materials and Methods

### 2.1. Materials

Polyvinylpyrrolidone with an average molecular weight of 360,000 (PVP-K90) was purchased from Spectrum Chemicals & Laboratory Products (New Brunswick, NJ, USA). 2-Hydroxypropyl beta cyclodextrin (HP-β-CD) with an average molecular weight of 1460 and d-α-tocopherol polyethylene glycol 1000 succinate (TPGS) were obtained from Sigma-Aldrich Inc. (St. Louis, MO, USA). Microcrystalline cellulose (Avicel PH 101) was purchased from Fluka (Hach Lange, Ireland). Croscarmellose sodium (Ac-Di-Sol) was obtained from FMC BioPolymer (Philadelphia, PA, USA). Talc powder was procured from Whittaker Clark & Daniels (South Plainfield, NJ, USA). Eudragit L100-55 was a kind gift from Rohm Pharma, GmbH (Weiterstadt, Germany). Magnesium stearate was procured from Winlab Laboratory Chemicals and Reagents (Leicestershire, UK). Lactose monohydrate spray-dried powder was purchased from Spectrum Chemical Mfg. Corp. (Gardena, CA, USA). Methanol was obtained from BDH Laboratory Reagents (Poole, England). 

### 2.2. Synthesis of PP10, the Acid Form of PP10 (PP10-Acid), and Its Sodium Salt (PP10-NaSalt)

The procedure for synthesizing PP10 was previously published by our group [18]. PP10-acid and PP10-NaSalt were prepared as below in Scheme 1. A solution of aqueous LiOH (3.48 mmol in 5 mL deionized H_2_O) was added to a solution of PP10 (1.74 mmol) in 5 mL methanol. The mixture was refluxed for 2 h at 80 °C, and the completion of the reaction was monitored by TLC (9:1 DCM/MeOH). Subsequently, the reaction mixture was neutralized with 2N HCl to pH 7. The formed PP10-acid solid was filtered and washed with water, then recrystallized from MeOH to yield 0.17 g (35.7%) of white solid.

PP10-acid was converted to PP10-NaSalt by adding NaOH (1 eq) to a solution of the acid in MeOH. The reaction was stirred for 1–2 h until PP10-NaSalt precipitate was formed, which was then filtered and washed with MeOH.

### 2.3. Stability of PP10 at Different pHs

The parent PP10 could hydrolyze into the corresponding PP10-acid either chemically (acid/base hydrolysis) or by the action of esterase in the blood, which could affect the PK and/or PD properties. This potential hydrolysis behavior was studied by investigating the stability of the parent methylester compound in buffer at different pHs, and in blood. To examine potential chemical degradation, a solution of PP10 (2 mg/mL) was prepared in methanol. Known volume (1 mL) of this methanol solution was diluted with a definite volume (20 mL) of buffer at pH 1.2, 5.5, or 7.4, corresponding to the gastric pH, the pH of the proximal small intestine, and the pH of the terminal ileum or physiological body fluids, respectively. Methanolic drug solution of the same concentration was also prepared as a reference. The different pH mixtures were incubated in a water bath at 37 °C overnight. The high-performance liquid chromatography (HPLC) chromatogram of the parent compound (methanolic drug solution) was compared with the corresponding drug buffer samples. The molecular weight of degraded product(s) was investigated using MASS scan. Details of the apparatus and conditions are described below.

For potential esterase enzymatic hydrolysis, PP10 was administered to male Sprague-Dawley rats. Whole blood samples were withdrawn over 24 h. The blood samples were spiked with a definite concentration of a PP10 standard solution. The parent PP10 and the potential PP10-acid metabolite were extracted, and the prepared samples were analyzed using HPLC. Detailed description of the extraction and HPLC methods will be illustrated in the pharmacokinetics section below.

### 2.4. In Vitro Antisickling, Hb Modification, and Hb Oxygen Equilibrium Studies of PP10 and Its Acid Analog with Sickle Whole Blood

The parent PP10, its acid metabolite PP10-acid, and the sodium salt PP10-NaSalt were tested using homozygous sickle cell (SS) blood for their pharmacologic activities. Leftover blood samples from individuals with SS were obtained (based on an approved IRB protocol at the Children’s Hospital of Philadelphia, with informed consent) and used for in vitro antisickling, Hb modification, and oxygen equilibrium studies following methods previously published by our group [25,26]. For the antisickling study, SS blood samples (hematocrit 20%) were incubated under air in the absence or presence of PP10, PP10-acid, or PP10-NaSalt (0.5 mM, 1 mM, and 2 mM) at 37 °C for 1 h, followed by incubation of the mixture under hypoxic conditions (2.5% O_2_/97.5% N_2_) at 37 °C for 2 h. Aliquots were fixed with 2% glutaraldehyde solution in sodium cacodylate buffer, pH 7.4, without exposure to air, and then subjected to microscopic morphological analysis of bright field images (at 40× magnification) of single layer cells on an Olympus BX40 microscope fitted with an Infinity 2 camera and coupled with Image Capture cellSens software (Olympus, Tokyo, Japan). 

Residual samples from the above study were washed in phosphate-buffered saline and hemolyzed by hypotonic lysis in deionized water for the oxygen equilibrium and Hb modification (Hb adduct formation) studies. Clarified lysates from the hemolyzed samples were subjected to cation-exchange HPLC (Hitachi D-7000 Series, Hitachi Instruments, Inc., San Jose, CA, USA), using a weak cation-exchange column (Poly CAT A: 30 mm × 4.6 mm, Poly LC, Inc., Columbia, MD, USA). Hemoglobin isotype peaks were eluted with a linear gradient of phase B from 0% to 80% at 410 nm (Mobile Phase A: 20 mM Bis-Tris, 2 mM KCN, pH 6.95; Phase B: 20 mM Bis-Tris, 2 mM KCN, 0.2 M sodium chloride, pH 6.55). A commercial standard consisting of approximately equal amounts of composite HbF, HbA, HbS, and HbC (Helena Laboratories, Beaumont, TX, USA) was utilized as reference isotypes. The areas of new peaks, representing HbS adducts, were obtained, calculated as percentages of total Hb area, and reported as levels of modified Hb or Hb adduct. For the oxygen equilibrium studies, a ~100 μL aliquot of the clarified lysate was added to 4 mL of 0.1M potassium phosphate buffer, pH 7.0, in cuvettes and subjected to hemoximetry analysis using the Hemox Analyzer (TCS Scientific Corp., New Hope, PA, USA) to assess Hb O_2_ affinity shift or P_50_, which is the partial pressure of oxygen (PO_2_) at which 50% of Hb is saturated with oxygen (SO_2_). Degree of Hb O_2_ affinity shift (ΔP_50_) was expressed as percentage of control DMSO-treated samples.

### 2.5. Preparation of Polymeric-PP10 Binary Systems

PP10 solid dispersions with PVP-K90 and Eudragit L100-55 were prepared in a drug to polymer ratio of 1:1 *w*/*w*. A drug inclusion complex with HP-β-CD was also prepared in a drug to polymer molar ratio of 1:1. Briefly, the calculated amount of drug and the studied polymer were dissolved in a sufficient amount of methanol over a magnetic stirrer until a homogenous mixture was obtained. The prepared mixture was poured into a porcelain dish and transferred into a hot air oven at 40 °C until the methanol was completely evaporated. The dried mass was crushed, passed through sieve no. 100, and finally stored in a desiccator until further characterization.

### 2.6. Solubility Study of PP10 and the Polymeric-PP10 Binary Systems

The equilibrium solubility of pure PP10 and the three polymeric drug binary systems with PVP-K90, Eudragit L100-55, or HP-β-CD were studied to investigate the effect of the complexation on the aqueous solubility of PP10. An excess quantity of the pure drug or the prepared binary systems was added to 5 mL of distilled water in a screw-capped vial and transferred to a thermostatically controlled shaking water bath (Model 1031; GFL Corporation, Burgwedel, Germany) at 25 ± 0.5 °C for 72 h. Aliquots from each vial were withdrawn and assayed for drug content every day spectrophotometrically at 275 nm using a Shimadzu UV-2600 UV–Vis spectrophotometer (Shimadzu Corporation, Kyoto, Japan) until the drug solubility values on two consecutive days were almost constant (i.e., did not change by more than 5%). Drug stock solution was prepared in methanol and different standard drug solutions in the range of 2–10 µg/mL were prepared. The absorbance of the prepared standard solutions was measured, resulting in a correlation coefficient (R^2^) of 0.9986.

### 2.7. Physicochemical Characterization of the Polymeric-PP10 Binary System

#### 2.7.1. Differential Scanning Calorimetry (DSC)

The thermal behavior of PP10, PVP-K90, HP-β-CD, and their polymeric-PP10 binary systems were investigated using a Shimadzu DSC TA-50 ESI DSC apparatus (Tokyo, Japan) calibrated with indium. Analysis was performed from 10 °C to 350 °C with a 10 °C/min heating rate.

#### 2.7.2. Fourier Transform Infrared (FT-IR) Spectroscopy

The FT-IR spectra of PP10, PVP-K90, HP-β-CD, and their polymeric-PP10 binary systems were collected using a Nicolet iS10 (Thermo Fisher Scientific, Waltham, MA, USA) within the wave number region of 4000–5000 cm^−1^.

#### 2.7.3. X-ray Powder Diffraction (XRPD)

To investigate the change in the crystalline state of PP10 after development of the polymeric-PP10 binary systems, the diffraction patterns of PP10 and the prepared binary systems with PVP-K90 and HP-β-CD were recorded at a scan speed of 0.5000°/min using a powder X-ray diffractometer (D/max 2500, Rigaku, Tokyo, Japan).

### 2.8. Formulation and Characterization of PP10 Oral Tablets

To develop an oral solid dosage form of PP10, the following tablet excipients were added to either the pure drug or the polymeric-PP10 binary systems with PVP-K90 or HP-β-CD; Avicel PH-101 (35% *w*/*w*) and spray-dried lactose monohydrate (15% *w*/*w*) were used as filler-binders; croscarmellose sodium “Ac-Di-Sol” (5% *w*/*w*) was added as a disintegrant; talc (0.5% *w*/*w*) and magnesium stearate (0.5% *w*/*w*) were used as glidant and lubricant. All ingredients were first passed through a United States standard sieve no. 30. A calculated weight of either the pure drug or the binary system was placed in a porcelain mortar and triturated well with the specified weight of Avicel PH-101 using a pestle. Lactose and Ac-Di-Sol were subsequently added, and trituration was continued until a homogenous powder mixture was obtained. The mixture was tumbled in a polyethylene bag for 10 min. Talc and magnesium stearate were finally added to the polyethylene bag and the powder mixture was further blended for 5 min.

Tablets of three different formulations containing either the pure drug, or the drug binary systems with PVP-K90 or HP-β-CD, were prepared by direct compression using a single station press (Erweka, Frankfurt, Germany) with a 9 mm circular, standard punch and die set. The mass of all tablets was kept constant, and each tablet was prepared to contain 30 mg of PP10.

The prepared tablets were evaluated for drug content, weight, thickness, hardness, friability, and in vitro disintegration time according to specifications of the United States Pharmacopeia 28/NF23 [27]. Drug content (*n* = 5) was determined by extracting the drug in methanol and assaying the methanolic extract for PP10 spectrophotometrically at 275 nm. Tablet weight (*n* = 10) was determined using a Mettler Toledo balance (Mettler Toledo, Switzerland). Tablet thickness (*n* = 6) was estimated using a digital micrometer (Mitutoyo Co., Kawasaki, Japan). Tablet hardness was assessed using TBH 200 hardness tester (Erweka, Hainburg, Germany). Tablet friability was evaluated using PTF10ER Friabilator (PharmaTest, Hainburg, Germany). The in vitro disintegration time was estimated using PTZ3 disintegration test apparatus (PharmaTest, Hainburg, Germany).

The in vitro dissolution of PP10 from the prepared tablets (*n* = 3) was determined using USP dissolution tester II (paddle type) DT 700 LH (Erweka, Hainburg, Germany). The dissolution was carried out in 900 mL of buffer at pH 1.2, 5.5, or 7.4. The paddle was operated at 75 rpm and equilibrated at 37 ± 0.5 °C. Five milliliters were withdrawn at predetermined time intervals of 0.25, 0.5, 1, 1.5, 2, 3, 4, 5, and 6 h and replaced with fresh and preheated dissolution medium at 37 ± 0.5 °C. The withdrawn samples were filtered and analyzed using the HPLC method described below.

### 2.9. Preparation and Characterization of PP10 Intravenous Formulation

Two different techniques, namely saturated polymeric drug solution [28,29] and TPGS aqueous micelle solution [16], were investigated to develop an intravenous drug preparation of PP10. In the first technique, different polymeric solutions of HP-β-CD (10–40 µM) in a hydroalcoholic solution (methanol:water, 50:50) were prepared. An excess amount of the drug was added to each solution. Each mixture was left overnight on a magnetic stirrer until complete evaporation of the organic solvent (methanol). Finally, an aliquot from each preparation was filtered using a Chromafil CA-20/25-S cellulose acetate 0.2 µm syringe filter. In the second technique, PP10-loaded TPGS micelle formulation was prepared by dissolving calculated amounts of TPGS (3% *w*/*v*) and PP10 (0.1% *w*/*v*) in 20 mL of methanol. Distilled water (20 mL) was added, and the organic solvent was completely evaporated using a Buchi Rotavapor R-200 (BÜCHI Labortechnik AG, Flawil, Switzerland). The resulting micellar dispersion was filtered through a Chromafil CA-20/25-S cellulose acetate 0.2 µm syringe filter.

The drug content in the filtrates was determined spectrophotometrically after dilution with methanol, against a methanol blank. Particle size of the PP10-TPGS micellar dispersion was assessed, in triplicate, using a Malvern Zetasizer Nano ZSP (Malvern Panalytical Ltd., Malvern, UK). Dynamic light scattering with noninvasive backscatter optics was utilized in the particle size measurement.

### 2.10. Pharmacokinetics Study

A single-dose one-period parallel design was utilized to investigate the pharmacokinetics of pure PP10, the polymeric-PP10 binary systems with PVP-K90 and HP-β-CD, and the PP10-TPGS aqueous micelle solution formulation. The study was conducted according to the guidelines of Good Clinical Practice (GCP), the International Conference on Harmonization (ICH), and the European Medicines Agency (EMA). The protocol for the animal study received prior approval from the Research Ethics Committee, Faculty of Pharmacy, King Abdulaziz University, Saudi Arabia (reference no. PH-1442-68) in 04/03/2021.

Male Sprague-Dawley rats (300–350 g) were used. The animals were maintained on a 12 h light/dark cycle (lights-on at 7 a.m.) with free access to food and water in a temperature-controlled room. After 7 days of acclimatization, animals were classified into four groups (*n* = 6). Group I was administered oral drug tablets containing pure drug (positive control). Group II was given oral drug tablets containing PP10-PVP K90 binary system. Group III was administered oral drug tablets containing PP10-HP-β-CD binary system. Group IV received the intravenous formulation containing PP10-loaded TPGS micelle. The tablets were crushed and suspended in a 1% carboxymethyl cellulose solution to develop an oral drug suspension suitable for oral administration by a gastric tube. The IV formulation was administered through the lateral tail vein. Animals were administered an oral dose of 100 mg/kg and an IV dose of 5 mg/kg.

Blood samples of 0.3 mL were collected from the jugular vein using a cannula into EDTA tubes before and after administration of the formulations at 0, 0.25, 0.5, 1, 2, 4, 8, 12, 24, 36, 48, and 72 h. Known volume (50 µL) of sterile water was added to an equal volume of each of the collected blood samples in a sterile vial to reach a volume of 100 µL. The vials were incubated for 10 min at 60–65 °C and a specified volume (200 µL) of 0.1 formic acid in acetonitrile was added. Vials were vortexed vigorously for 30 s, incubated for 30 min at 60–65 °C, and centrifuged at 5000 rpm for 20 min. A specified volume (150 µL) of the clear supernatant was diluted with an equal volume of sterile water, and 30 µL was injected into HPLC-PDA to estimate the drug concentration.

The concentration of the parent PP10 and PP10-acid metabolite in the whole blood samples were determined using a gradient elution HPLC chromatographic condition using a Waters Symmetry Shield C18 (Waters, MA, USA), 5 µm 3.9 × 150 mm column, and a Waters 2998 PDA detector (Waters, MA, USA) with detection wavelength adjusted to 274 nm. A gradient 0.1% formic acid and acetonitrile elution was adjusted for a total run time of 18.5 min. The injection volume was 30 µL and the flow rate was adjusted to 1 mL/min.

A standard calibration curve of the PP10-acid metabolite was constructed after hydrolysis of the parent PP10 compound with a diluent mixture of HCl, NaOH, and methanol. Standard acidic PP10 solutions in the range of 0.986–83.84 µg/mL were prepared after spiking the rat whole blood with a stock solution of the hydrolyzed acidic PP10. The parent ester compound (PP10) was used as an internal standard (IS). Since PP10 is a novel molecule that hydrolyses into the acid form, we used the ester as an IS to ensure accuracy of the analyte concentration measurement. The metabolite and IS were extracted as previously described. The calibration curve was constructed by plotting the concentration versus the area under the curve. The constants obtained were 1660.8, 905.1, and 0.998 for the slope, intercept, and the correlation coefficient (R^2^), respectively. For calculation of the metabolite concentration in the whole blood samples, the following equation was used:$$\mathrm{The}\;\mathrm{unknown}\;\mathrm{concentration}=\frac{Area\;of\;unknown-Intercept}{Slope}$$

Absolute recoveries of the metabolite and the IS in rat whole blood were calculated and found to be 90–110% and 100%, respectively. Quality control samples within the range of 0.9–90 µg/mL were prepared and the inter- and intraday precisions were found to be less than 5% of the relative standard deviation. The lower limit of quantification was 100 ng/mL.

The following pharmacokinetic parameters were calculated using PKsolver (an add-in program for pharmacokinetic data): maximum PP10-acid metabolite plasma concentration (C_max_), the time to reach the maximum plasma concentration (T_max_), the area under the plasma concentration–time curve from time zero until the last measurable drug concentration (AUC_0–t_), the area under the plasma concentration–time curve from time zero to infinity (AUC_0–inf_), the area under the moment curve from time zero to the end (AUMC), the mean residence time (MRT), the elimination half-life (t½), the elimination rate constant (K) which was calculated as 0.693/K, the total body clearance (Cl), and the apparent volume of distribution (Vd). The relative bioavailability for the tested tablets was determined by: AUC for binary system-based tablet/AUC for the positive control tablets ×100. Absolute bioavailabilities for the tablets in the binary systems were also determined.

### 2.11. Statistical Analysis

Data from the pharmacokinetic study were represented as mean ± SD and statistically analyzed using GraphPad Prism software, version 8 (GraphPad Inc., La Jolla, CA, USA). Significant differences among the rat groups were determined using a two-way analysis of variance (ANOVA) followed by Tukey’s multiple comparisons test. A *p*-value of less than 0.05 was considered statistically significant.

## 3. Results and Discussion

### 3.1. PP10 Hydrolyzed to the Acid Form (PP10-Acid) at Different Physiological pHs

During our previous in vitro study with whole blood [18], it became apparent that about 20% of the parent PP10 hydrolyzed into the PP10-acid metabolite. Although in this study PP10 was formulated as the parent ester, it is clear that in vivo the ester might hydrolyze into the acid form. We therefore investigated PP10 stability in vitro in straight methanol, and methanol buffered to pH 1.5, 5.5, or 7.4, corresponding to the pHs of gastric, the proximal small intestine, and the terminal ileum or physiological body fluid environments, respectively. We also investigated PP10 stability in vivo with blood obtained from PP10-dosed rats (see below). The HPLC chromatograms of PP10 in straight methanol and in the different methanol pH buffered solutions following overnight incubation at 37 °C are illustrated in Figure 1A. The peak height and area under the curve of the methanol-prepared PP10 sample decreased at pH 5.5 and pH 7.4, and an even more pronounced decrease was observed at pH 1.2. The MASS scan of PP10 in the different buffered samples revealed conversion of PP10 into the corresponding PP10-acid as illustrated in Figure 1B,C. In summary, PP10 appears to be stable in straight methanol but degrades into the acid form at pH 5.5 and pH 7.4. However, the most significant degradation occurred at pH 1.2, which was expected due to the acidic conditions obviously accelerating the hydrolysis of the ester to the acid form. Thus, depending on where the ester is primarily absorbed in the GI tract, the ratio of the acid and ester concentration in the blood may differ.

### 3.2. PP10 Binary System with HP-β-CD Showed Improved Solubility

The aqueous solubility of crystalline PP10 at 25 °C is very low at 0.012 ± 0.002 mg/mL, which prompted us to formulate the compound to improve its aqueous solubility. Inclusion complex with HP-β-CD and solid dispersions with PVPK90 and Eudragit L100-55 were prepared using the solvent evaporation technique. Results of the solubility study indicated only slight enhancement in the PP10 aqueous solubility upon formation of solid dispersions with PVPK90 (0.017 ± 0.0017 mg/mL) and Eudragit L100-55 (0.013 ± 0.0011 mg/mL), but a considerable increase in the aqueous solubility was achieved upon inclusion complexation with HP-β-CD (0.061 ± 0.0004 mg/mL).

Drug inclusion complex with cyclodextrins has attracted attention due to the hydrophilic nature of these carriers and their ability to form a stable complex with the hydrophobic guest molecule [30]. The inclusion complex is suggested to increase the aqueous drug solubility and dissolution rate, thereby enhancing diffusion across the biological membranes without changing the molecular structure or permeability characteristics [31,32]. Several studies show enhancement in the solubility, dissolution rate, and pharmacological activity of different drugs upon inclusion complexation with HP-β-CD [30,33,34,35,36]. The observed enhancement of aqueous solubility of PP10 upon complexation with HP-β-CD is attributed to inclusion of the drug in the hydrophobic cavity of the polymer, improving the wetting properties, changing the drug crystallinity, and decreasing the drug particles’ aggregation [16].

### 3.3. Physicochemical Characterization of the Binary Systems

Since only the binary system with HP-β-CD and, to a lesser extent, with PVPK90 showed improvement in the solubility of PP10, these two PP10-binary systems were selected for further physicochemical characterization. The differential scanning calorimetry DSC thermograms of the PP10-PVPK90 solid dispersion and PP10-HP-β-CD inclusion complex revealed an endothermic peak at 148.82 °C that represents the drug melting (data not shown). In comparison, the pure drug, and neat polymers PVPK90 and HP-β-CD melt at 148.82 °C, 114.57 °C, and 116.16 °C, respectively. Maximum endothermic peaks at around 143–147 °C and a broadening in the peak of the binary systems in this region indicated complexation of the drug with the polymers. The presence of the PP10 endothermic peak at the same temperature is a good indication of the stability of PP10 in the binary system formulations.

The FT-IR spectrum of PP10 depicted the characteristic drug carbonyl group (C=O) at 1720–1750 cm^−1^ and the drug ether (C–O) group at 1200–1300 cm^−1^ (Figure 2). The spectra of the drug binary systems with the polymers PVPK90 and HP-β-CD exhibited the same characteristic peaks, which confirm the preservation of the structural backbone of PP10 and its compatibility with the studied polymers.

The powder X-ray diffractogram of PP10 displayed the presence of sharp diffraction peaks which is an indication of the crystalline nature of the drug (Figure 3). The diffractograms of the binary systems were different from that of pure drug, which demonstrated peaks of lower intensity, disappearance, and formation of some new peaks. These observations indicate partial conversion of PP10 in the binary systems from crystalline to amorphous state.

### 3.4. Development of PP10 Oral Tablets

To investigate the quality attributes of the PP10-binary system oral tablet and its potential breakdown to the acid form, PP10-acid at different physiological pHs, drug tablets with pure PP10, and the HP-β-CD and PVP-K90 PP10-binary systems were further studied and characterized. Physical inspection of the tablets revealed no signs of sticking, picking, capping, or lamination. The tablet machine punches were clean with no signs of defects during compression. Quality control tests of the tablets (Table 1) revealed a uniformity in content and weight. Drug contents were 104.49 ± 3.71% for the pure PP10, and 99.48 ± 6.94% and 102.34 ± 6.71% for the HP-β-CD and PVP-K90 binary systems, respectively. The weight of tablets containing pure drug and PVPK90 were 138 ± 0.01 and 137 ± 0.02 mg, respectively, while that of HP-β-CD was 281 ± 0.02 mg. Friability of all tablets was less than 1%. Drug thickness was in the range 1.72 ± 0.03–3.32 ± 0.06 mm. The hardness of tablets containing pure drug and PVPK90 was 49.67 ± 5.7 and 84.67 ± 7.8 N, respectively, while that of HP-β-CD was 204.33 ± 8.08 N. Tablet disintegration time was less than 10 min in all of the prepared formulations, but was higher in tablets containing HP-β-CD (8.16 ± 0.08 min) than in those containing pure drug (5.19 ± 0.13 min) and PVPK90 (5.84 ± 0.22 min). In general, the tablet containing PP10-HP-β-CD inclusion complex was of greater weight due to the high content of HP-β-CD utilized in the development of the binary system in a 1:1 drug to polymer molar ratio—an effect that was reflected in the tablet thickness, hardness, and disintegration time.

The in vitro dissolution of PP10 from the tablets was studied at different pHs. Results revealed marked hydrolysis of the ester from all the prepared tablets at pH 1.2 and pH 5.5, and partially at pH 7.4. Notably, the release profile at pH 7.4 illustrated significant improvement in the drug dissolution, especially for tablets containing the drug inclusion complex with HP-β-CD due to solubility enhancement (data not shown). This finding is in good agreement with the results for the drug stability study discussed above that suggest the potential hydrolysis of PP10 into the acid at the various physiological pHs of the gut. Enhancement of PP10 solubility following complexation with HP-β-CD resulted in better in vitro drug dissolution, as previously reported for vardenafil tablets containing a drug-HP-β-CD inclusion complex [37]. The authors attributed this enhancement in the in vitro drug dissolution to the improvement of drug solubility following inclusion complexation with HP-β-CD.

### 3.5. PP10 Showed Improved Intravenous Formulation with TPGS Micelle

Intravenous (IV) drug administration requires the drug formulation to be in the form of an aqueous solution. IV administration of poorly water-soluble drugs represents a major challenge, prompting development of different approaches to develop IV formulations suitable for poorly water-soluble drugs. Salt formation, co-solvency, micelle formation, complexation, and preparation of sub-micrometer lipid-based systems, such as emulsions and liposomes, are common techniques to prepare such IV formulations for poorly soluble drugs [38]. In this study, two different techniques were used to develop PP10 IV formulations containing the highest drug concentration: saturated polymeric drug solution with HP-β-CD, and aqueous micelle solution with TPGS. Both techniques have been successfully used by our research teams to develop aqueous solutions for poorly water soluble drugs, such as glimepiride [29], simvastatin [28], and vinpocetine [16]. HP-β-CD polymeric solutions of 10, 20, 30, and 40 µM polymer concentration showed aqueous dose-dependent drug solubility of 0.0535 ± 0.006, 0.0811 ± 0.004, 0.116 ± 0.001, and 0.121 ± 0.009 mg/mL, respectively. The TPGS aqueous micelle solution formulation showed a drug concentration of 0.502 ± 0.01 mg/mL and a particle size of 26 ± 3 nm. Accordingly, this formulation was selected as an IV dosage form for further study.

### 3.6. PP10 Formulation with HP-β-CD Binary System Showed Enhanced PK Properties and Bioavailability

Our study led to significant improvement in PP10 solubility for oral delivery using HP-β-CD binary system, and IV delivery using TPGS aqueous micelle solution. These formulations, in addition to the pure drug and the marginally improved formulation with PVPK90, were used to test PP10 pharmacokinetic properties in vivo using rats. RBCs are the target site of action for aromatic aldehydes in the treatment of sickle cell disease. Only about 0.5–15% of these drugs remain in plasma and most partition into the RBC [18,19,21,22]. However, the exact distribution is affected by several factors, including hematocrit, plasma protein binding, RBC partitioning, and metabolism. Plasma levels are thus not a robust marker of aromatic aldehyde at the target site and would be much more prone to error. Whole blood levels are therefore a much better predictor of aromatic aldehyde delivery into the body to target Hb and correlate quite well to RBC levels. Accordingly, the concentration of PP10 was estimated in the whole blood.

The pharmacokinetic profile (Figure 4) of PP10 in whole blood was constructed after administration of 100 mg/kg of the tablets to the rats. The calculated pharmacokinetic parameters are presented in Table 2. As expected, animals administered tablets containing the binary mixture of PP10 and HP-β-CD showed superior results when compared with groups administered tablets containing PP10 and PVPK90 solid dispersion or pure drug. Maximum PP10 (PP10-acid) blood contents of 53.71 ± 9.51, 58.488 ± 10.06, and 93.3 ± 22.6 μg/mL were observed after 8.67 ± 7.76, 3.33 ± 1.03, and 6.67 ± 2.07 h for tablets containing pure drug, PP10-PVPK90, and PP10-HP-β-CD binary systems, respectively. No parent PP10 compound (PP10) was observed in any of the analyses, suggesting complete hydrolysis into the acid form, PP10-acid. Tablets containing PP10-HP-β-CD binary system demonstrated higher AUC and AUMC when compared with the other studied groups. Additionally, tablets containing PP10-HP-β-CD binary system showed a relative bioavailability of 173.37% when compared with the pure drug tablets, while tablets containing PP10-PVPK90 exhibited a relative bioavailability of 88.81% compared with the pure drug tablets. The absolute bioavailability of the tablet containing PP10/HP-B-CD complex was calculated and found to be 106.34%.

In general, an enhanced pharmacokinetic parameter and bioavailability of PP10 were observed for PP10-HP-β-CD binary system, which confirms the efficiency of drug complexation with HP-β-CD that was also reflected in the drug solubility and the in vitro release profile.

### 3.7. Both Parent PP10 and the Acid Form (PP10-Acid) Exhibit Biological Activity

It is clear that the parent PP10 compound can be hydrolyzed into the acid metabolite, PP10-acid, as observed in vitro in whole blood (~20%) [18], as well as in the stability studies with both the formulated and non-formulated binary systems, and the in vivo PK studies. This therefore begs the question as to whether the acid form, like the ester, will be biologically active. To answer this question, we incubated PP10, PP10-acid, and PP10-NaSalt with sickle whole blood, and assayed for their pharmacological effect, including Hb adduct formation, Hb oxygen affinity, and RBC sickling inhibition. The result, as shown in Figure 5, suggests that both the ester and the free acid forms show dose-dependent antisickling effect, although the former exhibits a stronger effect. For example, at 2 mM concentration, PP10 showed 36% more RBC sickling inhibition than PP10-acid. Interestingly, PP10-NaSalt showed almost no activity, likely due to inability of the highly ionic PP10 to traverse the RBC membrane and partition into the RBC to bind Hb. We also believe that the reduced biological activity of the more polar PP10-acid, compared with the less polar PP10, is due to slow RBC partitioning by the former. Based on the co-crystal structure of Hb with PP10, [18] we expect PP10-acid to have similar protein interactions, but we cannot discount possible differences. As expected from the trend in the antisickling activity, PP10 showed the highest Hb adduct formation, followed by PP10-acid, with significantly less adduct formation by PP10-NaSalt. What is most interesting, and intriguing, is that even though PP10-acid showed lower RBC sickling inhibition and Hb adduct formation when compared with PP10, we observed a reverse dose-dependent potency in Hb oxygen affinity. For example, at 2 mM concentration, PP10-acid increased Hb oxygen affinity by about 30% when compared with the ester. This observation is quite unusual since increased Hb oxygen affinity normally correlates with sickling inhibition. The only plausible explanation, in the absence of PP10-co-crystal, is that the acid moiety leads to better stabilization of the R-state Hb, which does not necessarily contribute to the O_2_-dependent antisickling effect like the parent ester.

## 4. Conclusions

PP10 was chemically synthesized as a prodrug for the treatment of sickle cell disease and potentially other inherited blood disorders with underlying hypoxia condition. Although the pure drug showed significant potency in vitro, the solubility is too low to allow for any significant GI solubility or oral bioavailability. In this study, the aqueous drug solubility was enhanced following inclusion complexation with HP-β-CD and, to a lesser extent, solid dispersion with PVPK90. Oral tablets of PP10 were successfully developed and showed acceptable quality attributes. Aqueous PP10 IV formulation was prepared with TPGS utilizing the micelle solubilization technique. Tablets contain PP10-HP-β-CD inclusion complex showed superior pharmacokinetic profile and bioavailability when compared with the corresponding tablets loaded with either PP10-PVPK90 solid dispersion or pure drug. PP10 is currently undergoing detailed in vivo pharmacodynamic studies which will be followed by an IND—enabling studies for the treatment of SCD.

## Data Availability

Not applicable.

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
