# Peer review of "Improving the Solubility and Oral Bioavailability of a Novel Aromatic Aldehyde Antisickling Agent (PP10) for the Treatment of Sickle Cell Disease"

_pharmaceutics, 2021, doi:10.3390/pharmaceutics13081148_

Round 1

Reviewer 1 Report

The manuscript titled ‘Improving the Solubility and Oral Bioavailability of a Novel..’ By Tarek Ahmed is a novel study.

Although study is good, manuscript doenot look focused. Authors have not written the manuscript in simple way. Long sentences should be avoided and the text should be simpliefied for the easy understanding of the readers. Overall the manuscript is very confusing.

  1. References are not written in MDPI style.
  2. Introduction – first paragraph should be shortened, Focus on the solubility aspect and not on the disease as the paper deals with formulation.
  3. Introduction should specify the solubility of PP10. As authors already knew that compound has poor solubility and needs solubilisation.
  4. Looking at the title of the paper, authors should delet the synthesis section (section 2.2, 2.3) Abstract doesnot mention anything about sysnthesis of PP10 and its esters.
  5. If authors were known that pp10 has antisickling effect, why they have tested it again? Section 2.5.
  6. Line 101. Either use pp10 as the abbreviation or pp10-ester. Using 2 names for 1 novel novel therapeutic agent for sickle cell disease is misleading.
  7. TPGS 3% w/v is extremely high concentration. Authors need to check its CMC. Even oral safe permissible limits are much lower.
  8. Figures are going outside the margin of page
  9. Pharmacokinetic data after intravenous injection is missing. Authors should add it in graphs of tablets. No need of separate figure.
  10. Line 379. Using pp10 ester as IS isnot justifiable.
  11. Figure 5- What the error bars indicate SD or SEM?

Author Response

The manuscript titled ‘Improving the Solubility and Oral Bioavailability of a Novel..’ By Tarek Ahmed is a novel study.

Although study is good, manuscript doenot look focused. Authors have not written the manuscript in simple way. Long sentences should be avoided and the text should be simpliefied for the easy understanding of the readers. Overall the manuscript is very confusing.

1. References are not written in MDPI style.

Response: References have been modified to meet the journal style.

2. Introduction – first paragraph should be shortened, Focus on the solubility aspect and not on the disease as the paper deals with formulation.

Response: The introduction has been modified. The first paragraph has been shortened. Details about the drug solubility has been added.

3. Introduction should specify the solubility of PP10. As authors already knew that compound has poor solubility and needs solubilisation.

Response: The solubility of PP10 has been mentioned in the introduction section, paragraph 3.

 4. Looking at the title of the paper, authors should delet the synthesis section (section 2.2, 2.3) Abstract doesnot mention anything about sysnthesis of PP10 and its esters.

Response: The synthesis of PP10, which was previously published, has been removed from the manuscript as suggested by the reviewer.

5. If authors were known that pp10 has antisickling effect, why they have tested it again? Section 2.5.

Response: PP10 was tested in this study as a comparative study with the acid and NaSalt analogs.

6. Line 101. Either use pp10 as the abbreviation or pp10-ester. Using 2 names for 1 novel novel therapeutic agent for sickle cell disease is misleading.

Response: We highly appreciate the reviewer comment. PP10 has been used.

7. TPGS 3% w/v is extremely high concentration. Authors need to check its CMC. Even oral safe permissible limits are much lower.

Response: We completely agree with the reviewer point of view in that the CMC for TPGS is much less than 3%. During development of the IV formulation, we have tried to use lower concentration of TPGS but unfortunately no significance enhancement in the drug solubility was obtained. Also, the prepared PP10-loaded TPGS micelle formulation was unstable at lower concentration. Kim et al develop an oral formulation of sirolimus using a TPGS micellar solution at a concentration of 50 mg/mL (5%).    

Min-Soo Kim, Jung-Soo Kim, Won Kyung Cho & Sung-Joo Hwang. Enhanced solubility and oral absorption of sirolimus using D-α-tocopheryl polyethylene glycol succinate micelles. Artificial Cells, Nanomedicine, and Biotechnology, 2013; 41: 85–91

8. Figures are going outside the margin of page.

Response: All figures have been adjusted.

9. Pharmacokinetic data after intravenous injection is missing. Authors should add it in graphs of tablets. No need of separate figure.

Response: In this work, we have focused on the development of an oral PP10 formulation. The intravenous injection was used to calculate the bioavailability as represented in table 2. A separate work for development of an intravenous formulation, that achieves an aqueos drug solubility in the range of 2-5 mg/ml, utilizing pH adjustment, derivatization, co-solvent, micelles, complexation, nano-formulation and emulsion techniques are currently running and will be submitted as a separate work. 

 10. Line 379. Using pp10 ester as IS is not justifiable.

Response: PP10 is a novel molecule that hydrolyses into the acid form. We have used the ester as an internal standard to ensure accuracy of the analyte concentration measurement.  This explanation has been added to the revised manuscript.  

11. Figure 5- What the error bars indicate SD or SEM?

Response: The error bars indicate SD. Figure 5 has been modified in the revised manuscript.

Reviewer 2 Report

Reviewer’s Comments:

  Ahmed et al present an interesting manuscript -"Improving the Solubility and Oral Bioavailability of a Novel Antisickling Agent, PP10 for the Treatment of Sickle Cell Disease" that demonstrates the development of oral and parenteral formulations to improve PP10 solubility and bioavailability.

Comments:

  1. Abstract: Results section- Are the values Mean ± SEM or Mean ± SD?
  2. Please include the catalog numbers of the kits and reagents used in this study.
  3. Please add a brief paragraph on “future directions to this study” at the end of the discussion/conclusions section.
  4. Please be consistent with the style of references especially with respect to the page numbers and DOI.

Author Response

Ahmed et al present an interesting manuscript -"Improving the Solubility and Oral Bioavailability of a Novel Antisickling Agent, PP10 for the Treatment of Sickle Cell Disease" that demonstrates the development of oral and parenteral formulations to improve PP10 solubility and bioavailability.

Comments:

1. Abstract: Results section- Are the values Mean ± SEM or Mean ± SD?

Response: All the values are expressed as mean ± SD. For more clarification to the readers, this explanation has been explained in the tables and figures.

2. Please include the catalog numbers of the kits and reagents used in this study.

Response: The synthesis of PP10, which was previously published, has been removed from the manuscript as suggested by the reviewer. Accordingly the reagents that used in this part have been removed. Other materials used in this study have been fully described in the materials section.

3. Please add a brief paragraph on “future directions to this study” at the end of the discussion/conclusions section.

Response: PP10 is currently undergoing detailed in vivo pharmacodynamic studies, which will be followed by an IND-enabling studies for the treatment of SCD. This explanation has been added to the conclusion section.

4. Please be consistent with the style of references especially with respect to the page numbers and DOI.

Response: References have been modified to meet the journal style.

Reviewer 3 Report

The paper entitled “Improving the Solubility and Oral Bioavailability of a Novel  1  Antisickling Agent, PP10 for the Treatment of Sickle Cell Dis- 2  ease” by Ahmed et al. is in interesting attempt of practically important alteration of dissolution profiles of some therapeutic agents for sickle cell disease. They used complex formulations in the solid form and utilizing well established technology of intrusion of hydrophobic drug into a hydrophilic carrier. The concept of the project is clearly defined, methodology well documented and results of sound scientific values. The paper deserve publishing after minor revisions due to some misfortune ways of documenting of obtained results.

It seems to me that Authors were in a hurry and manuscript has several technical problem. Here are some issues:

Although in general the paper is well prepared however version I have received for reviewing has several technical problems as for example bad quality and incomplete figures (cut off from page view). In general description of figures is not clear and requires editing. For example on Fig.1,4 units are unknown, Fig.1 is unclear with unrecognizable numbers, units and plots.

The manuscript does not utilize Journal template consequently. There are different fonts size and typefaces, interlines, etc. For example title of paragraph 2.4 utilizes other style than 2.3.

Schemes of synthesis and equations not numbered.

I do not know if PP10 is commonly accepted symbol and using in the abstract is not very informative.

The corresponding author provides private e-mail. Wouldn’t it to be better contacting officially?

Some sentences are unclear: line 101: … “Our group using structure-based efforts identified…”

Line 476: 1/cm, rather 1/Cm

Line 274: frequencies are not expressed in a wave length units

Generally citation belong to line, in which they were used and dots should be placed after not before (for example lines 55, 62, …).

Digits numbers of data presented in Table 2 seem to be odd. What was the significance level?

Some abbreviations collected in the end of manuscript seem to be odd, as for example br, h, rt, etc. but TPGS and other less obvious are missing.

I am impressed by amount of works in their multistage study, which was necessary for providing the final results contained in fig.5 and 6. Splendid work! Please improve the way of presentation for letting reader appreciate your work.

Author Response

The paper entitled “Improving the Solubility and Oral Bioavailability of a Novel  1  Antisickling Agent, PP10 for the Treatment of Sickle Cell Dis- 2  ease” by Ahmed et al. is in interesting attempt of practically important alteration of dissolution profiles of some therapeutic agents for sickle cell disease. They used complex formulations in the solid form and utilizing well established technology of intrusion of hydrophobic drug into a hydrophilic carrier. The concept of the project is clearly defined, methodology well documented and results of sound scientific values. The paper deserve publishing after minor revisions due to some misfortune ways of documenting of obtained results.

  • It seems to me that Authors were in a hurry and manuscript has several technical problem. Here are some issues: Although in general the paper is well prepared however version I have received for reviewing has several technical problems as for example bad quality and incomplete figures (cut off from page view). In general description of figures is not clear and requires editing. For example on Fig.1,4 units are unknown, Fig.1 is unclear with unrecognizable numbers, units and plots.

Response: More clear figures have been used in the modified manuscript. Bad quality figures, such as figure 2 has been removed.

  • The manuscript does not utilize Journal template consequently. There are different fonts size and typefaces, interlines, etc. For example title of paragraph 2.4 utilizes other style than 2.3. Schemes of synthesis and equations not numbered.

Response: The journal template has been used in the modified version. Upon the reviewer comments, the procedure for synthesizing PP10 and the scheme that was used to make the PP10-acid and PP10-NaSalt have been rewritten.  

  • I do not know if PP10 is commonly accepted symbol and using in the abstract is not very informative.

Response: PP10 is an antisickling aromatic aldehyde, and the full name is Methyl 6-((2-formyl-3-hydroxyphenoxy)methyl)picolinate, which we have now provided in the abstract.

  • The corresponding author provides private e-mail. Wouldn’t it to be better contacting officially?

Response: Only the official “university” e-mail has been used in the modified manuscript.

  • Some sentences are unclear: line 101: … “Our group using structure-based efforts identified…”

Response: We have revised the sentence.

  • Line 476: 1/cm, rather 1/Cm

Response: The label for the x-axis in figure 3 “currently amended as figure 2” has been corrected.

  • Line 274: frequencies are not expressed in a wave length units

Response: The sentence has been corrected in the modified manuscript.

  • Generally citation belong to line, in which they were used and dots should be placed after not before (for example lines 55, 62, …).

Response: Citation of the references has been corrected.

  • Digits numbers of data presented in Table 2 seem to be odd. What was the significance level?

Response: Data presented in table 2 are represent as the mean±SD. We have reviewed all the numbers and corrected some typo mistakes.

  • Some abbreviations collected in the end of manuscript seem to be odd, as for example br, h, rt, etc. but TPGS and other less obvious are missing.

Response: The abbreviations section has been removed and acronyms have been fully described at their first presence in the modified manuscript.

  • I am impressed by amount of works in their multistage study, which was necessary for providing the final results contained in fig.5 and 6. Splendid work! Please improve the way of presentation for letting reader appreciate your work.

Response: we highly appreciate the reviewer comments. The manuscript has been modified to present data in a good way.

Round 2

Reviewer 1 Report

No comments.